# Skin parasite landscape determines host infectiousness in visceral leishmaniasis

Johannes S.P. Doehl[1], Zoe Bright [2,4], Shoumit Dey [1,2], Helen Davies[1,2], John Magson[3], Najmeeyah Brown[1], Audrey Romano[1], Jane E. Dalton[1], Ana I. Pinto[1], Jon W. Pitchford[2] & Paul M. Kaye [1]

Increasing evidence suggests that the infectiousness of patients for the sand fly vector of visceral leishmaniasis is linked to parasites found in the skin. Using a murine model that supports extensive skin infection with *Leishmania donovani*, spatial analyses at macro-(quantitative PCR) and micro-(confocal microscopy) scales indicate that parasite distribution is markedly skewed. Mathematical models accounting for this heterogeneity demonstrate that while a patchy distribution reduces the expected number of sand flies acquiring parasites, it increases the infection load for sand flies feeding on a patch, increasing their potential for onward transmission. Models representing patchiness at both macro- and micro-scales provide the best fit with experimental sand fly feeding data, pointing to the importance of the skin parasite landscape as a predictor of host infectiousness. Our analysis highlights the skin as a critical site to consider when assessing treatment efficacy, transmission competence and the impact of visceral leishmaniasis elimination campaigns.

[1] Centre for Immunology and Infection, Department of Biology and Hull York Medical School, University of York, York YO10 5DD, UK. [2] Departments of Biology and Mathematics, University of York, York YO10 5YW, UK. [3] Fera Science Ltd (Fera), Applied Innovation Campus, Sand Hutton, York YO41 1LB, UK. [4] Present address: EPSRC Centre for Predictive Modelling in Healthcare, University of Exeter, Exeter EX4 4QF, UK. Correspondence and requests for materials should be addressed to J.W.P. (email: jon.pitchford@york.ac.uk) or to P.M.K. (email: paul.kaye@york.ac.uk)

Kinetoplastid parasites of the *Leishmania donovani* complex are the causative agents of zoonotic and anthroponotic visceral leishmaniasis (VL), associated with 3.3 million disability-adjusted life years[1] and 20,000–40,000 annual deaths in 56 affected countries[2]. There is no available prophylactic vaccine[3], available drugs are either too expensive or may have significant toxicity issues[4]; and while the incidence of VL in South Asia is falling, evidence for the contribution of active vector control is limited.

All mammalian-infective *Leishmania* species have a digenetic life-cycle, occurring as intracellular amastigotes in the mammalian host and as extracellular promastigotes in the insect vector[5]. Infective metacyclic promastigotes are regurgitated into the skin of a mammalian host by the bite of female hematophagous phlebotomine sand flies (Diptera: Psychodidae: Phlebotominae)[6]. *L. donovani* disseminates from the skin to deeper tissues, like the spleen, liver and bone marrow, causing systemic pathology, but the skin is commonly asymptomatic[7]. Blood parasite load (parasitemia) has been considered the main determinant of outward transmission potential and as a surrogate measure of

host infectiousness in VL[8]. Therefore, mathematical models of VL epidemiology have been considering VL hosts as homogeneous infection units, assuming each blood meal taken from a host to be equally infective to a sand fly[9]. However, in vivo data correlating parasitemia and vector infection success are scarce and conflicting[10, 11].

In contrast, recent studies using sand flies for parasite detection (xenodiagnosis) in dogs with canine VL have shown that skin parasite load correlates well with parasite uptake by sand flies, suggesting that the skin plays an important role in host infectiousness[11–13]. It is important to note that sand flies are telmophages, feeding from blood-pools caused by skin-laceration, rather than solenophages, like mosquitos, which feed by directly piercing blood vessels. This distinction is important, because, mechanistically, telmophagy allows parasite uptake from the skin as well as blood. In contrast to blood, skin has a greater potential for generating heterogeneity in parasite distribution, and both dispersal and dispersion of parasites within the skin is linked to that of the host myeloid cells in which these parasites reside. Pertaining to concepts of landscape epidemiology[14], host cells

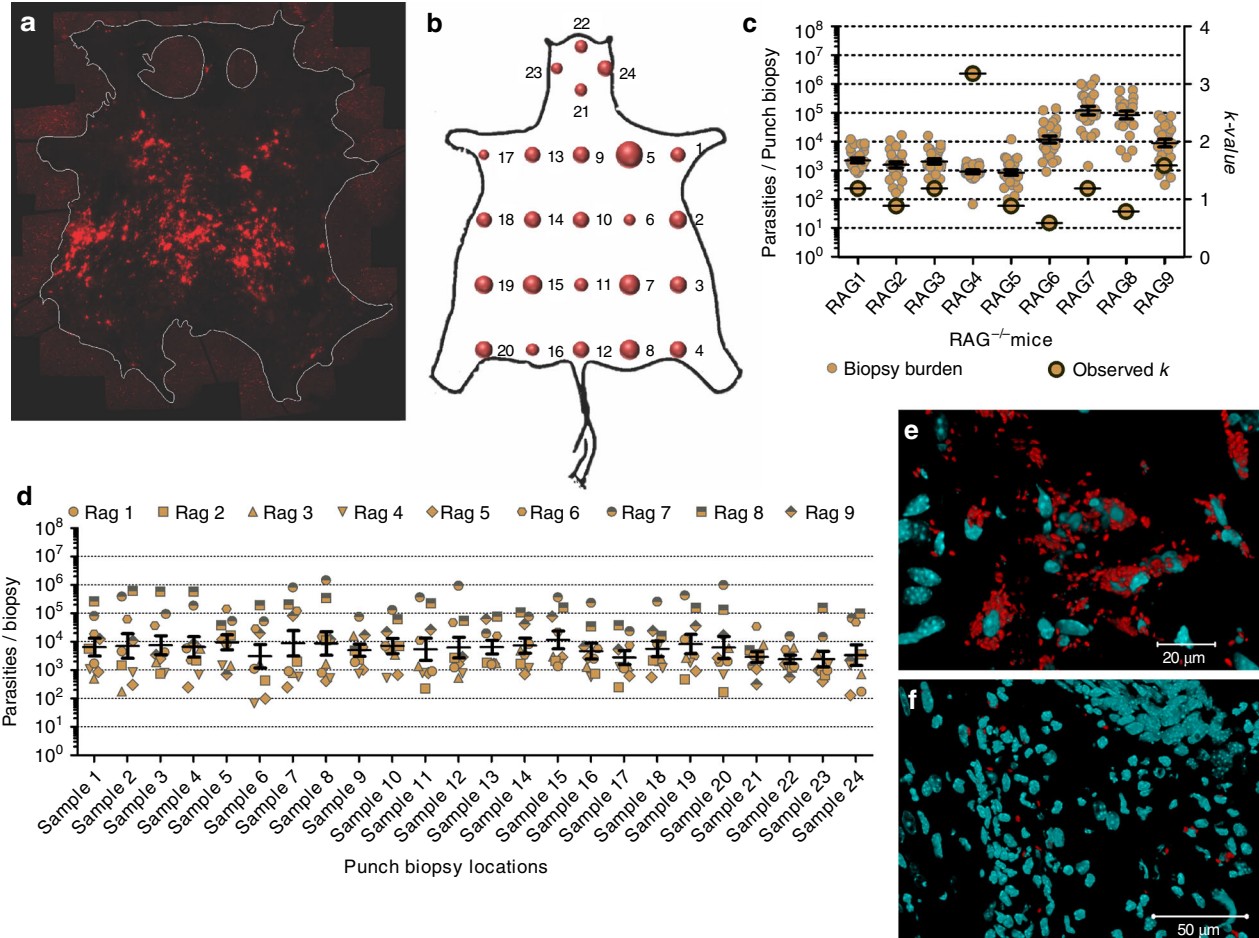

**Fig. 1** Analyzing skin parasite load and distribution. **a** Composite image of tdTom-expressing *L. donovani* parasites (*red*) in the skin of a representative infected RAG mouse (RAG9), imaged from the hypodermal face by fluorescent stereomicroscopy (see Supplementary Fig. 1 for more details). Skin edges were marked for better contrast. **b** Bubble plot of accumulative biopsy data for all RAG mice (biological replicates $N = 9$; see Supplementary Fig. 2 for more details). **c** Scatter plot of parasite loads per skin biopsy for each mouse (ø 0.8 cm; vol. 12 mm$^3$; technical repeats $N = 24$ per mouse; *left y-axis*) and observed $k$-value for parasite distribution for each mouse (*right y-axis*). Mean and standard errors are shown for parasite load data. Skin-biopsy parasite load variability per mouse were analyzed by two-tailed one-sample $t$-test ($P < 0.001$ each), data variance between all mice (biological replicates $N = 9$) within the group by Brown-Forsythe test ($P < 0.001$) and mean biopsy parasite load by one-way Kruskal-Wallis test ($P < 0.001$). **d** Scatter plot for parasite density per biopsy location for all 24 biopsy locations across all mice (biological replicates $N = 9$ per biopsy site, total $N = 216$) analyzed by one-way Kruskal-Wallis test ($P = 0.996$). Mean and standard error bars are shown. **e, f** Magnified z-stack tile-scan images of skin areas containing heavily **e** and sparsely **f** parasitized mononuclear cells (full image in Supplementary Figs. 4, 5, respectively). Scale bars: 20 μm in **e**, 50 μm in **f**

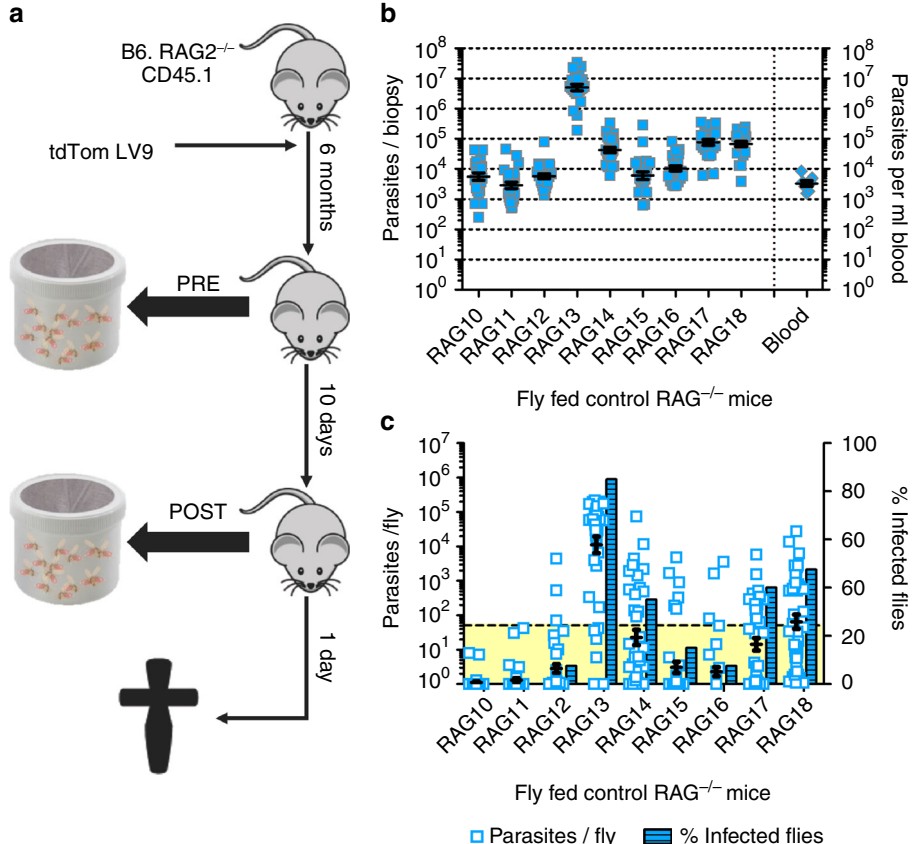

**Fig. 2** Analyzing impact of skin parasite burden on outward transmission. **a** Schematic representation of the experimental sand fly exposure procedure. **b** Scatter plot of parasite loads in skin for each mouse (ø 0.8 cm; vol. 12 mm³; technical repeats $N = 24$ biopsies per mouse; total $N = 216$) and blood parasitemia (per ml; biological replicates $N = 9$) of each mouse. Mean and standard error bars are shown. Skin parasite load variability per mouse was analyzed by two-tailed one-sample $t$-test ($P < 0.001$ each), data variance between all mice (biological replicates $N = 9$) within the group by Brown-Forsythe test ($P < 0.001$) and mean biopsy parasite load by one-way Kruskal-Wallis test ($P < 0.001$). **c** Experimental qPCR data of parasite loads per sand fly after the second mouse exposure (scatter plot; technical replicates $N = 40$ per mouse, total $N = 360$) paired with the ratio (%) of sand flies harbouring >50 parasites (threshold area in *yellow*) at day 7 PBM (bar chart). Mean and standard error bars are shown (Scatter plot only). Sand fly parasite load variability between Fly Fed control mice (biological replicates $N = 9$) by Brown-Forsythe test ($P < 0.001$) and mean sand fly parasite burden by one-way Kruskal-Wallis test ($P < 0.001$)

may be subject to cellular, molecular and physiological conditions in the skin that determine and restrict their spatial localization, and hence that of the parasites they contain, resulting in a three-dimensional heterogeneous micro-landscape of parasite density and distribution in the host skin. Potentially, this skin parasite landscape may generate significant variability that might have an impact on transmission. However, quantitative analysis of this potential heterogeneity and its impact on infectiousness has not previously been performed.

Here, we describe an outward transmission model of VL involving tandem dimer Tomato (tdTom) *L. donovani*-infected B6.CD45.1.$Rag2^{-/-}$ (herein referred to as RAG) mice and *Lutzomyia longipalpis*, and its use in investigating skin parasite load and distribution (landscape) as potential predictors of host infectiousness. Using this infection model, we found a significant parasite load in the skin of long-term infected RAG mice. Moreover, these showed a heterogeneous distribution in the skin that was strongly correlated to parasite uptake by feeding sand flies. To further understand these determinants of host infectiousness, we developed mathematical models derived from ecology to investigate the involvement of skin parasite load and spatial heterogeneity on outward transmission potential. Our data clearly demonstrate that the heterogeneous landscape of parasites in skin is an important determinant of host infectiousness.

## Results

**L. donovani distribute heterogeneously in host skin.** We infected immunodeficient RAG mice with tdTom-*L. donovani* parasites by intravenous inoculation through the lateral tail vein to induce parasite dissemination to the viscera and secondary skin infection (as opposed to primary skin infection due to sand fly bite). We then imaged the body skin (excluding ears, snout and paws) of ~6-month infected RAG mice ($N = 9$) and found randomly distributed patches of parasites of variable density (Fig. 1a and Supplementary Fig. 1). Parasite distribution was quantified on a macro-scale (24 punch biopsies of ø 0.8 cm; vol. ~12 mm³) by qPCR for *Leishmania* kinetoplastid minicircle DNA (kDNA; Fig. 1b and Supplementary Fig. 2). Our data showed significant variance of parasite load across biopsies in each mouse (two-tailed one-sample $t$-test: $P < 0.001$ each), ranging from 66 to ~$1.5 \times 10^6$ parasites (Fig. 1c and Supplementary Fig. 2), in keeping with imaging data (Fig. 1a and Supplementary Fig. 1). The variance in biopsy parasite load also showed significant differences between mice (Brown-Forsythe test: $P < 0.001$; Fig. 1c). Mean biopsy parasite loads also significantly varied between mice (one-way Kruskal-Wallis test: $P < 0.001$), ranging from ~$10^3$ (mouse: RAG4) to ~$3 \times 10^5$ (mouse: RAG7) parasites per biopsy (Fig. 1c). Across mice, however, there was no indication of preferential sites for parasite

accumulation (one-way Kruskal-Wallis test: $P = 0.996$; Fig. 1d and Supplementary Fig. 3A). Taking volume into account, the skin parasite load was ~17 (RAG5) to ~4423 (RAG7) times higher than blood parasitemia (calculated per $cm^3$; Supplementary Fig. 3B), suggesting that host skin provides a potentially greater source of parasites for sand fly infection than blood (two-tailed Wilcoxon signed-ranks test: $P = 0.008$).

Since parasite uptake from the skin by sand fly bite is possible due to telmophagy, the observed heterogeneity (patchiness) in skin parasite distribution raised the question of whether this was important for determining infectiousness. Simple mathematical modeling illustrates this concept. Under conditions where parasites are distributed homogenously, all sand flies might be expected to acquire similar numbers of parasites wherever they feed (Supplementary Fig. 3C). Conversely, with a patchy distribution, many sand flies might be expected to acquire no or negligible numbers of parasites, but those feeding at a "patch" would acquire larger numbers, depending on patch density (Supplementary Fig. 3D). To further investigate these concepts using empirically-derived data, skin parasite counts were analyzed by maximum likelihood estimation to characterize the distribution, using Akaike Information Criterion[15] to avoid over-fitting (Supplementary Table 1). At the macro-scale, the skin parasite data (obtained by qPCR) was best fitted by a negative binomial (NB) distribution, which is defined by the mean ($\mu$; mean biopsy parasite load per mouse) and a dispersion factor ($k$). The overall macro-scale parasite distribution can be categorized as homogenous ($k \to \infty$), geometric ($k = 1$) or heterogeneous (patchy; $k \to 0$). Put simply, for small values of $k$ patchiness is increased and the number of infected sand flies is decreased, but the number of parasites ingested by those sand flies feeding on a patch is increased (Supplementary Fig. 3D). In silico, the effect of $k$ on host infectiousness can be predicted by adjusting the experimentally observed $k$ (Fig. 1c) either towards zero ($k \to 0$) or towards infinity ($k \to \infty$). Therefore, we developed a mathematical model that used the empirical skin parasite load data fitted to a NB distribution, a maximum feed volume of 1.6 $mm^3$ for *Lu. longipalpis* and an infection threshold of 1,000 amastigotes (derivation of the infection threshold is discussed further below). Adjusting $k$ in this model predicted that patchiness enhances host infectiousness for low mean skin-biopsy parasite loads ($\mu = \sim 2 \times 10^3$ parasites; RAG2 and 3; Supplementary Fig. 3E), while the inverse was true for higher mean skin-parasite loads ($\mu > 10^4$ parasites; RAG6-9; Supplementary Fig. 3E). By quantifying heterogeneity, $k$ can help to explain why sand fly infection success can increase where patches harbour parasites at higher density for a given mean load, as compared to a homogeneous distribution. Considering that parasite loads in asymptomatic VL hosts are often at the lower end, heterogeneity may be very important in the natural transmission cycle of *L. donovani*.

A more detailed analysis requires a quantification of heterogeneity at smaller (micro) scales. We imaged cryo-preserved 10 µm thick cross-sections of tdTom-*L. donovani*-infected mouse skin by z-stack tile-scanning confocal microscopy. Parasites were primarily observed in the dermis, rather than the epi- or hypodermis. Skin parasite patches corresponded to an accumulation of densely parasitized mononuclear cells at the side of the dermis distal to the epidermis (reticular dermis; Fig. 1e and Supplementary Fig. 4). Other dermal areas showed very sparsely parasitized cells, or were devoid of parasites altogether (Fig. 1f and Supplementary Fig. 5). To investigate parasite distribution, squares equivalent to the blood pool size were superimposed on confocal z-stack tile-scan images of skin cryo-sections and tdTom-*L. donovani* parasites were counted per square. The count data was fitted to candidate distributions as for the macro-scale. Again, a NB distribution was found to fit the data best, allowing the use of

$k_{Micro}$ to characterize parasite heterogeneity within the models in a way exactly analogous to the macro-scale (Supplementary Table 2). In summary, the data reveal that the skin parasite landscape is heterogeneous at both macro- and micro-scales.

**Parasite load and distribution correlate with infectiousness.** To test experimentally the importance of the skin for outward transmission, we exposed a new cohort of ~6-month infected RAG mice (Fly Fed control; $N = 9$) twice to female *Lu. longipalpis* sand flies ($N = 50$ per mouse per feed) 10 days apart (Fig. 2a). We exposed only the shaven right flanks to female sand flies for blood feeding (see Methods). The analysis of the skin parasite load was conducted by stereomicroscopy (Supplementary Fig. 6) and qPCR (Fig. 2b and Supplementary 7) as before. The variance in biopsy parasite loads was significantly different between mice within this group (Brown-Forsythe test: $P < 0.001$; Fig. 2b) as observed previously (Fig. 1c). The skin parasite distribution also remained patchy (Supplementary Figs 6 and 7), reflecting a general observation that the higher the mean parasite load on the skin of a mouse, the denser and more ubiquitous the parasite patches observed.

*Lu. longipalpis* is a widely-used permissive vector species supporting *L. donovani* persistent infection and development. Sand flies were maintained for 7 days post blood meal (PBM) to allow parasite establishment and expansion and then killed for quantitation of parasite kDNA by qPCR. Quantitation of parasite load per sand fly indicated that there was variability in the outward transmission potential between mice (Fig. 2c and Supplementary Fig. 8A). To estimate the number of infected sand flies that might develop mature, transmissible infections, we introduced an arbitrary threshold of 50 parasites per midgut, assuming that sand flies harbouring <50 parasites at day 7 PBM will not develop mature, transmissible infections by day 12–14 PBM. Using this cutoff, the data indicate that the mice that infected most sand flies (Fig. 2c and Supplementary Fig. 8B) also produced higher mean parasite loads (Fig. 2c and Supplementary Fig. 8A). On average, we did not see a significant change in outward transmission potential due to repeat exposure in the Fly Fed control group (two-tailed Mann-Whitney test: $P = 0.126$; Supplementary Fig. 8C).

To further challenge our hypothesis, we employed two additional mouse cohorts. We selectively reconstituted immunity in these mice through adoptive $CD4^+$ or $CD8^+$ T-cell transfer 24 h after the first sand fly exposure, and hence 9 days prior to the second sand fly exposure (Fly Fed $CD4^+$ T-cell recipient [$N = 9$] and Fly Fed $CD8^+$ T-cell recipient [$N = 9$]; from three independent experiments; Supplementary Fig. 9A). T-cell transfer was performed to address the question of whether immune reconstitution could affect outward transmission and/or the skin parasite landscape. Based on qPCR analysis, adoptive $CD4^+$ and $CD8^+$ T-cell transfer did not significantly alter the systemic spleen and liver parasite loads compared to the Fly Fed control group (one-way Kruskal-Wallis test: spleen, $P = 0.647$; liver, $P = 0.214$; Supplementary Fig. 10A, B). The analysis of the skin parasite load was conducted by stereomicroscopy (Supplementary Figs. 11 and 13) and qPCR (Supplementary Figs. 9B, 9C, 12 and 14) as before. The Fly Fed $CD4^+$ T-cell recipient group showed a significantly higher mean skin-parasite load than the Fly Fed Control and Fly Fed $CD8^+$ T-cell recipient groups (one-way Kruskal-Wallis test: $P < 0.001$; Supplementary Fig. 9D). The Fly Fed control and Fly Fed $CD8^+$ T-cell recipient groups showed comparable mean skin-parasite loads to each other (one-way Kruskal-Wallis test: $P = 0.429$; Supplementary Fig. 9D). Again, the variance in biopsy parasite loads was significantly different between mice within each group (Brown-Forsythe test: $P < 0.001$ each; Supplementary Fig. 9B, C). However, the skin-parasite distribution remained

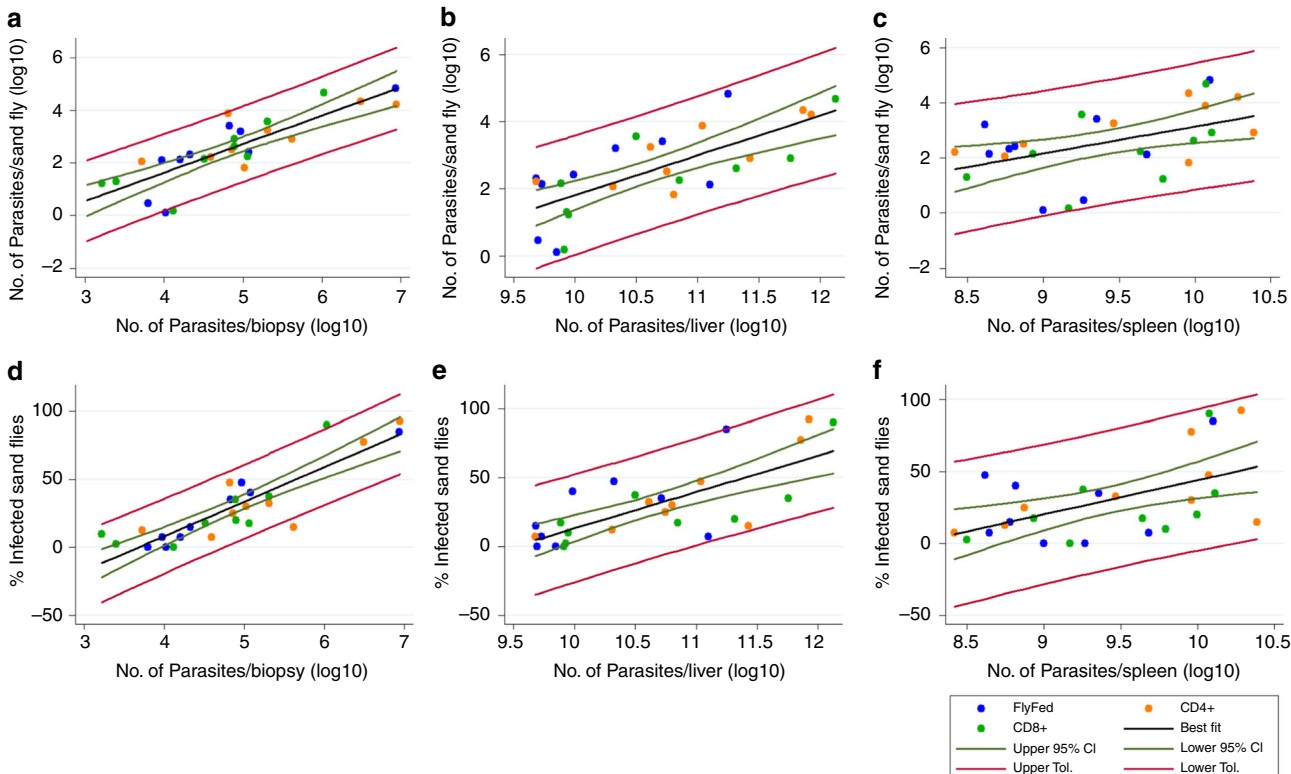

**Fig. 3** Correlating tissue parasite load to outward transmission success. Correlation plots comparing sand fly parasite load (determined by qPCR) with host parasite loads in **a** skin, **b** liver and **c** spleen (all determined by qPCR). Correlation plots comparing frequency of infected sand flies with host parasite loads **d** skin, **e** liver and **f** spleen (all determined by qPCR). Mice from all experimental groups were included in the analyses, but represented by group in the plots (see legend). The graphs show the line of best fit (*black*), the 95% confidence intervals (CI; *green*) and the tolerance bands (*red*). All qPCR data were log$_{10}$-transformed prior to analysis. Biological replicates $N = 9$ per group, total $N = 36$

patchy for mice of all groups (Supplementary Figs. 6, 7, 11-14). These results further indicated that the main difference in parasite distribution between mice from all groups was patch densities (Supplementary Figs. 1, 6, 11 and 13), which corresponded to average skin biopsy parasite loads per mouse (Figs. 1c and 2b and Supplementary Fig. 9B, C).

Looking at the outward transmission potential, the data strengthened our observation with the Fly Fed control group that the mice that infected most sand flies (Supplementary Fig. 8E, G) also produced higher mean parasite loads (Supplementary Fig. 8D, F and 9E, F). Further, it emerged that the Fly Fed CD4$^+$ T-cell recipient group was the most efficient in transmitting parasites 9 days post T-cell adoptive transfer (second sand fly exposure; one-way Kruskal-Wallis test: $P < 0.001$ and $P = 0.004$, vs. Fly Fed control and CD8$^+$ T-cell recipient groups, respectively), whereas there was no difference between Fly Fed control and CD8$^+$ T-cell recipient groups (one-way Kruskal-Wallis test: $P = 0.238$; Supplementary Fig. 8C). Interestingly, a significant increase in outward transmission potential was observed in the CD4$^+$ and CD8$^+$ T-cell recipient groups by comparing in repeat exposures pre- and post-adoptive transfer (first and second sand fly exposure, respectively; two-tailed Mann-Whitney $U$ test: $P = 0.035$ and $P < 0.001$, respectively; Supplementary Fig. 8C). Although not the focus of the current study, these data raise the intriguing possibility that immune reconstitution has a potentially detrimental effect in terms of infectiousness that warrants further investigation.

**Skin parasite load correlates with outward transmission.** We next ran a series of linear regression analyses of parasite load per

sand fly (qPCR), and numbers of infected sand flies of the second sand fly exposure (Fig. 2c and Supplementary Fig. 9E, F) against mouse skin, liver, spleen and blood parasite load. Residuals for all linear regressions were checked for normal distribution and uniform variance for test validity. We found that skin and liver parasite load (qPCR) were the best predictors of outward transmission potential ($P < 0.0001$; Fig. 3a, b, d, e). By extrapolation with regards to the skin, transmission potential would approach zero if skin parasite load was <2,000 parasites per biopsy, although this threshold may be lower with increased patchiness. Spleen parasite load was a more moderate predictor of parasite load per sand fly ($P = 0.012$; Fig. 3c) and number of infected sand flies ($P = 0.005$; Fig. 3f), independently of the means used to assess parasite load (Supplementary Fig. 15A, B, D, E). Parasitemia did not correlate with outward transmission ($P = 0.093$ and 0.22, respectively; Supplementary Fig. 15C, F). A multiple linear regression analysis containing the skin, liver, spleen and blood parasite loads as independent predictors of parasite load per sand fly and number of infected sand flies showed an adjusted $R^2$ of 0.785 and 0.908, respectively, and confirmed skin parasite load to be the best predictor of outward transmission potential (skin: $P < 0.001$ and $P < 0.001$; liver: $P = 0.034$ and $P = 0.005$; spleen: $P = 0.086$ and $P = 0.012$; blood: $P = 0.894$ and $P = 0.155$, respectively). These observations are in concordance with recent findings from Borja et al. in dogs[11]. However, Borja et al.[11] also reported a strong correlation of parasitemia to their xenodiagnostic data, which we could not confirm for our mouse model. This difference in correlation of parasitemia to outward transmission potential may be due to the different disease states of the analyzed animals. Potentially, more severely sick animals harbour more parasites in the blood,

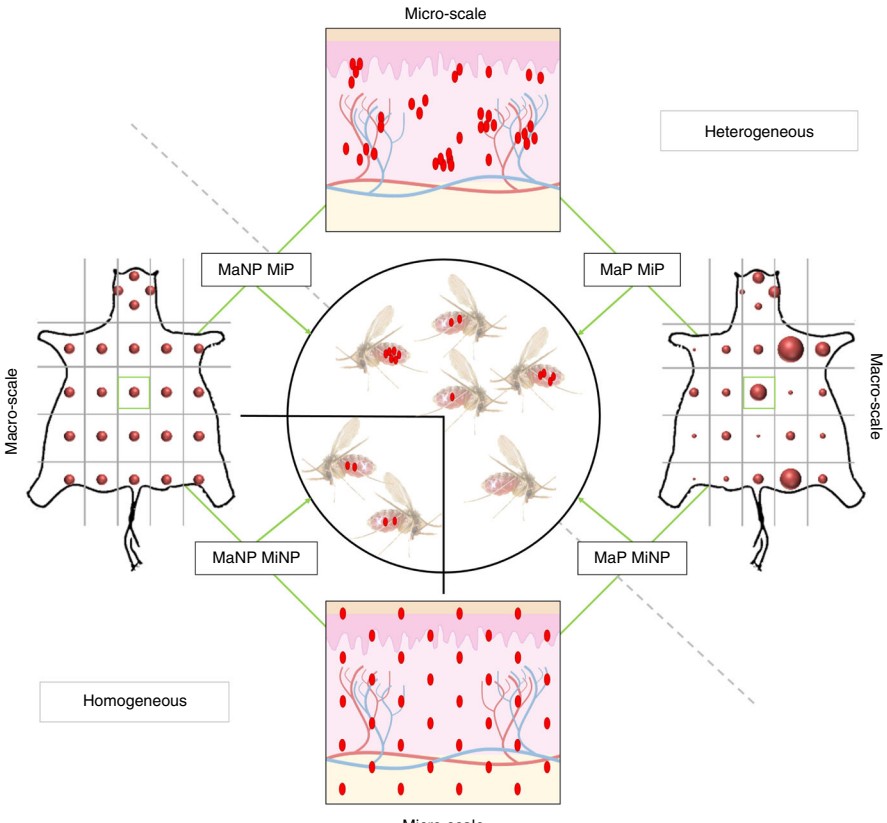

**Fig. 4** Schematic representation of transmission model. Parasite distributions are assumed to be either homogeneous (not patchy = NP) or heterogeneous (patchy = P) in the skin. The macro-scale (Ma) considers heterogeneity at the scale of a skin biopsy, based on stereomicroscopic analysis of the whole skin and qPCR of punch biopsies. Each biopsy in turn has a micro-scale (Mi) parasite distribution, which may be either homogeneous or heterogeneous. A macro- and micro-homogeneous distribution (MaNP MiNP) results in all sand flies ingesting similar amounts of parasites, patchiness at either scale (MaP MiNP and MaNP MiP) results in heterogeneity in parasite uptake between sand flies, and patchiness at both scales (MaP MiP; as observed in the data) maximizes the heterogeneity in parasite uptake

compared to less symptomatic animals, despite high systemic parasite loads. Either way, our study significantly adds to the growing evidence that skin parasite load is an important determinant of outward transmission.

**Modeling predicts the importance of parasite landscape.** Since we cannot control for parasite distribution in the host, we used the mathematical characterizations of patchiness and combined them with our empirical data into a simple mechanistic model of patchy feeding in heterogeneous environments (detailed in Supplementary Note 1)[16]; models accounting for such heterogeneity are known to improve predictions of disease spread as well as providing practical insight into control measures[17, 18]. A preliminary model based on Pitchford et al.[16] is detailed in Supplementary Note 2. Further, Fig. 4 shows a schematic outline of an evolved model approach, considering parasite load and distribution at both the macro- and micro-scale to predict outward transmission success (detailed in Supplementary Note 3).

Implementing the model requires several assumptions about the parameters driving outward transmission, such as minimum infection loads, blood pool and fly feed volumes (Supplementary Table 3). Although no definitive data on the minimum amastigote load required for successful sand fly infection is currently available, it is unlikely that a single amastigote is always sufficient to infect a sand fly[19–21]. Successful infection of *Phlebotomus duboscqi* was previously reported to occur in ~33% of artificially

fed and fully engorged sand flies ingesting >10 lesion-derived *L. major* amastigotes[22]. Conversely, Sádlová et al. (2016) required ingestion of ~600 *L. donovani* amastigotes for reliable infection of *Phlebotomus argentipes*[21]. Under natural sand fly feeding conditions, the infection threshold can be expected to be significantly higher for successful infection[23]. On modeling the infection threshold (Supplementary Fig. 16A), an infection threshold of 1000 amastigotes was found to correspond to ~33% of infection success and was therefore implemented in the model (see also Supplementary Note 4). For simplicity, blood pool (0.19 mm$^3$) and fly feed (1.6 mm$^3$) volumes were initially fixed (see Methods for volume definitions; Supplementary Table 3). By incorporating our empirical data from the Fly Fed control group of RAG mice, this model predicted 100% transmission for mice with $\mu > \sim 5 \times 10^4$ mean parasites per biopsy (RAG13, 14, 17, 18), assuming a homogeneous parasite distribution, whereas all other mice would not transmit (Fig. 5a). This model is akin to the situation where only a blood parasitemia, which is homogeneous by definition, is responsible for the outward transmission potential. Accounting for macro-scale patchiness reduced the predicted transmission potential of these mice (RAG13, 14, 17, 18), but additionally predicted enhanced transmission in mice with $\mu < 5 \times 10^4$ mean parasites per biopsy (Fig. 5a), a situation much more akin to data derived from experimental feeding experiments (Fig. 2c).

For the incorporation of micro-scale heterogeneity, we assumed the blood pool volume defined the area in the dermis that would contribute parasites to the blood meal. The model

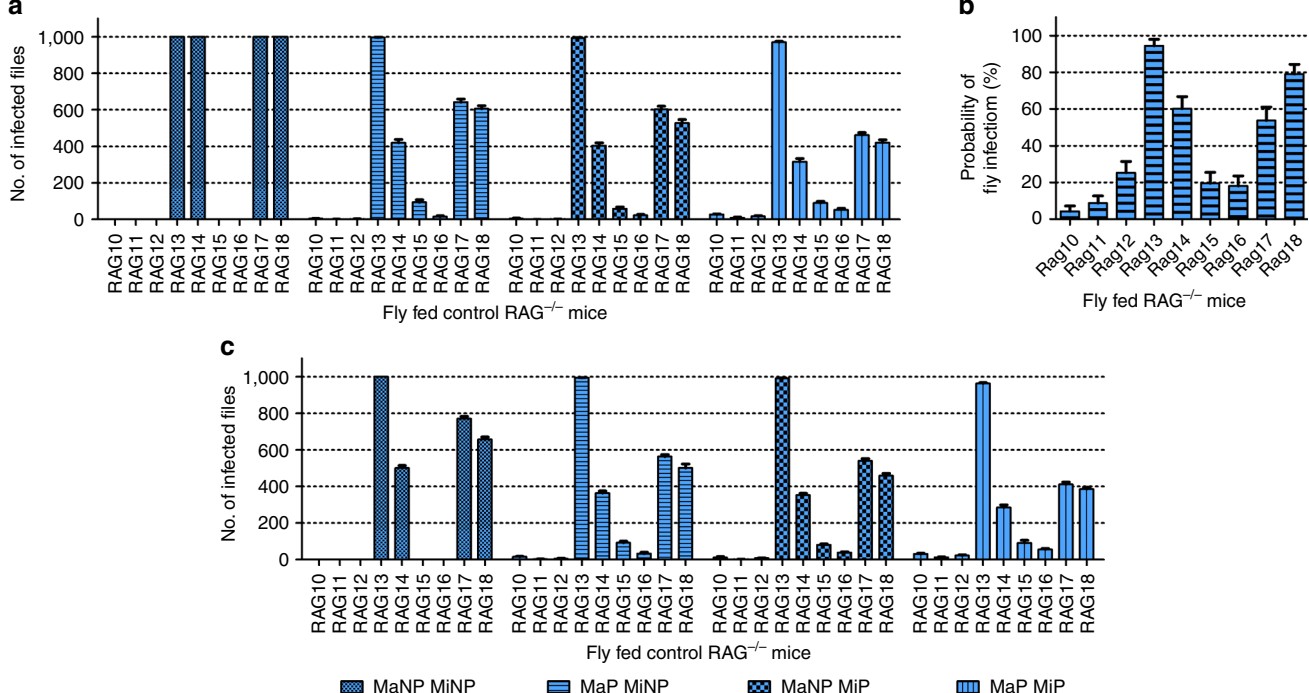

**Fig. 5** Predicting outward transmission potential by distribution. **a** Model predicting host infectiousness to sand flies using fixed parameters (infection threshold = 1000 amastigotes, blood pool volume = 0.19 mm$^3$, fly feed volume = 1.6 mm$^3$). Either a macro-(Ma)/micro-(Mi) scale homogeneous (MaNP MiNP; NP = non-patchy), a macro-scale patchy micro-scale homogeneous (MaP MiNP; P = patchy), a macro-scale homogeneous micro-scale patchy (MaNP MiP) or a macro-/micro-scale patchy (MaP MiP) distribution is assumed. Graphs are based on 10 model iterations. Standard error bars are shown. **b** Estimation of probability of sand fly infection based on experimental sand fly data. Graphs are based on 10 model iterations. Standard error bars are shown. **c** Model predicting host infectiousness with infection threshold reduced to 500 amastigotes, a 50% amastigote loss post ingestion and with uniformly random blood pool volumes (0.05–0.42 mm$^3$) and fly feed volumes (0.1–1.6 mm$^3$). Either a macro-(Ma)/micro-(Mi) scale homogeneous (MaNP MiNP; NP = non-patchy), a macro-scale patchy micro-scale homogeneous (MaP MiNP; P = patchy), a macro-scale homogeneous micro-scale patchy (MaNP MiP) or a macro-/micro-scale patchy (MaP MiP) distribution is assumed. Model outcomes are based on experimental results obtained from the Fly Fed control RAG mice as shown in Fig. 2b. See Supplementary Fig. 16 for a similar analysis using data from RAG mice receiving CD4$^+$ or CD8$^+$ T cells

showed that, similar to the observations at the macro-scale, micro-scale patchiness alone (Fig. 5a) can reflect experimental data (Fig. 2c) better than a homogeneous skin parasite distribution. A combined macro- and micro-scale patchy distribution showed the best fit to our experimental transmission data. To verify these results, we used the sand fly transmission data (Fig. 2c) to calculate the infection probability per mouse (Fig. 5b). The observed pattern is a good approximation to the experimental (Fig. 2c) and predicted data based on macro-/micro-scale patchiness (Fig. 5a). Hence, our data argue strongly that not only skin parasite load, but also heterogeneous parasite distribution within the skin are important predictors of host infectiousness.

To further test our model, we challenged the model with data derived from mice adoptively transferred with CD4$^+$ and CD8$^+$ T cells (CD4$^+$; $N = 9$ and CD8$^+$; $N = 9$). For mice receiving CD4$^+$ T cells, the macro-/micro-scale patchy model again was the best fit for the observed transmission data (2-way Friedman test with Wilcoxon signed-ranks test as post hoc: $P = 0.051$; other models $P < 0.05$; Supplementary Fig. 16B). Due to the significantly higher skin load, CD4$^+$ T-cell reconstituted RAG mice were predicted to be more infectious to sand flies than controls and this was again borne out in the experimental data (one-way Kruskal–Wallis test: $P < 0.001$; Supplementary Figs. 9E and 16B). The model verification, using the sand fly data, again showed a good approximation to the pattern of the experimental sand fly data (Supplementary Figs. 9E and 16C) as observed for the Fly Fed control group. Finally, using data from CD8$^+$ T-cell recipient

RAG mice showed highly consistent results (Supplementary Fig. 16E, F), adding further support to the assertion that skin parasite load and distribution have a major impact on infectiousness under these conditions.

**Model predictions are robust to parameter value variability**. Our model is deliberately parsimonious, driven by empirically available data on patchiness at macro- and micro-scales. Nevertheless, it was able to make experimentally verifiable predictions about outward transmission potential as a function of skin parasite load and distribution. Changing the assumptions relating to model parameters like blood pool and fly feed volumes necessarily changes the modeled outcomes quantitatively, but the qualitative predictions are consistent.

Our theoretical model can be readily adjusted and extended to accommodate e.g., changes in infection threshold (e.g., 10–1000 amastigotes), uniform random blood pool and sand fly feed volumes for individual sand flies (Supplementary Table 3). It has previously been reported that there is a significant loss of amastigotes (50–88%) post ingestion[19, 20], which we can also account for in our model. Here, we reduced the infection thresholds to 500 amastigotes, which is closer to previously observed amastigotes numbers (~600) required for reliable sand fly infection with *L. donovani* by artificial feeding[21]. Using a 500 amastigote infection threshold, a 50% parasite loss rate post ingestion, uniform random blood pool (0.05–0.42 mm$^3$) and fly feed volumes (0.1–1.6 mm$^3$), the model still favours a patchy

distribution to account for experimental data (Fig. 5c and Supplementary Figs. 16D, G). However, abandoning an infection threshold altogether in our model renders predictions that do not correspond to the range of observed infection probabilities (Supplementary Fig. 17; Supplementary Note 4). At the micro-scale, blood pool volume could arguably be substituted by sand fly proboscis volume (Supplementary Fig. 18); hence considering only parasites in the immediate bite area to be relevant for outward transmission and increasing the importance of blood parasitemia. Nevertheless, such a model still indicates the importance of skin parasite patchiness (Supplementary Fig. 19), though it becomes less accurate at reflecting experimental data compared to the blood pool micro-scale model. In addition, using a bootstrap method to avoid fitting the empirical skin parasite load data to a distribution showed the same variability and results in qualitatively identical predictions as previously, although at a coarser level (Supplementary Fig. 20; Supplementary Note 5). This showed that our conclusions did not rely on any extreme values within the fitted distributions.

## Discussion

Heterogeneity at the vector and host population level has also been shown by mathematical models to be important in the transmission cycle of other diseases borne by hematophagous insects[24–26]. In nature, not all potential hosts are equally exposed to vectors due to differences in habitat. Those hosts living closer to vector breeding ground are more likely to encounter vectors. In addition, host characteristics like age, sex, size, blood type and health status may affect host-attractiveness to vectors resulting in heterogeneous biting of hosts. Heterogeneous biting rates are also affected by vector and host spacing, peaking in areas of greatest vector density and in the periphery of greatest host density[26]. Interestingly, inclusion of heterogeneity into these models always resulted in basic reproductive rates of disease, $R_0$, equal to or greater than (but never less than) those found by assuming homogeneity[24, 25]. These concepts are directly transferable to skin parasite distribution and vector encounter rates of skin parasite patches. Therefore, it may not come as a surprise that heterogeneity in the distribution of Leishmania parasites in the host skin should enhance a host's outward transmission potential.

The results of our modeling also suggest that there may be upper and lower parasite loads within which patchiness contributes to infectiousness. For example, in some mice (e.g., RAG13 and RAG24) the introduction of patchiness has minimal impact on infectiousness because their mean skin-parasite load is so high that it approaches a highly dense homogeneous distribution ($k \rightarrow \infty$; Supplementary Figs. 6 and 11). Conversely, for other mice (RAG10 and RAG26) introduction of patchiness has a minimal effect because mean skin parasite load is so low that significant patches do not form (Supplementary Figs. 6 and 11).

The versatility and applicability of our model could be further increased in the future by including considerations of reproductive numbers per individual ($R_i$) rather than mean reproductive numbers ($R_0$) for VL[18], multi-host situations with varying transmission potentials per species[27], sand fly feeding behaviour and host preference[8], superspreaders[18, 28] and targeted vs. global intervention methods[18].

In conclusion, our patchy skin parasite model shows how the observed large variance and strong zero-bias in the experimental sand fly data is usefully explained by mean skin-parasite load ($\mu$) and the observed patchy skin-parasite distribution ($k \rightarrow 0$). Using skin parasite load and distribution to predict outward transmission potential is biologically relevant due to the pool feeding behaviour of sand flies. This has significant implications for the design and analysis of clinical studies. For xenodiagnosis studies aimed at identifying asymptomatic "transmitters" within a population exposed to L. donovani[29], we suggest that analysis of blood parasitemia needs to be augmented both by quantitative multiple skin-site analysis of skin parasite load (e.g., using qPCR or LAMP assays) and for distribution by histopathology. At a macro-scale, if our analysis of infected RAG mice holds true, then single-site analysis may have a high likelihood of misrepresenting infectiousness. For elimination programs, it is important to understand how drug treatment affects patchiness as well as parasite load in the skin. If dense parasite patches remain largely unaffected by drug treatment, then "cured" patients might remain as effective transmitters of disease. Conversely, if treatment was more selective for parasite patches (e.g., due to altered parasite metabolic activity at such sites), then single infecting amastigotes may remain in the skin and eventually re-form patches that restore infectiousness. The occurrence of patchiness in VL reservoirs may be an evolutionary means to enhance outward transmission potential without requiring high parasite loads; an attempt by the parasite to potentially mimic asymptomatic bite sites, which are normally associated with outward transmission potential in domestic reservoirs[13]. Finally, our analysis has potential implications for other vector-borne diseases, e.g., African trypanosomiasis[30] and Lyme disease[31], if pathogen distribution within the host is found to be similarly skewed with respect to transmission site.

## Methods

**Ethics statement.** All animal usage was approved by the University of York Animal Welfare and Ethics Review Committee and the Ethics Review Committee at FERA Science Ltd., and performed under UK Home Office license ('Immunity and Immunopathology of Leishmaniasis' Ref # PPL 60/4377).

**Mouse, Leishmania and sand fly lines.** C57BL/6 (B6; originally obtained from The Jackson Laboratory) and B6.CD45.1.$Rag2^{-/-}$ mice (originally obtained from the National Institute of Medical Research, London, UK) were used in this study. All animals were bred and maintained at the University of York according to UK Home Office guidelines. The Ethiopian Leishmania donovani strain (MHOM/ET/1967/HU3) had been transfected with a tandem dimer Tomato (tdTom) gene[32] to generate tdTom-L. donovani in another study[33]. The parasites were maintained in B6.CD45.1.$Rag2^{-/-}$ (RAG) mice by repeat-passage of amastigotes. Parasites were isolated from the spleens of infected mice as previously described[34]. Briefly, the spleens from infected RAG mice were harvested, homogenized in RPMI and debris pelleted down at $136 \times g$ and 37 °C. Mammalian cells were lysed in 25 mg Saponin per 20 ml RPMI, parasites pelleted and washed three times at $2040 \times g$ and 37 °C. Amastigote aggregates were separated by forcing the suspension 2–3 times through a 26-gauge needle. All mice used in this study were male and were infected at the age of 6–10 weeks via intravenous injection at the lateral tail vein with $\sim 8 \times 10^7$ tdTom-L. donovani amastigotes in 200 µl RPMI. Experimental mice were 30–40 weeks old at the end of the study. The colony of New World phlebotomine sand flies, Lu. longipalpis, is maintained at the insectarium of FERA Science Limited under contract with the University of York. Standard procedures used for sand fly colony husbandry were previously published[35].

**Mouse exposure to sand flies.** Mice were anaesthetized with ketamine (90 mg kg$^{-1}$) and dormitor (0.2 ml kg$^{-1}$) in PBS by intra-peritoneal injection. An area equivalent to 10–12 punch biopsies (ø 0.8 cm) was exposed on the right flank by shaving with electrical clippers (Supplementary Fig. 21) and individual mice were laid flat with the shaven flank onto individual sand fly holding cages. About 50 female Lu. longipalpis sand flies were allowed to feed on an anaesthetized mouse for 30–40 min at room temperature (~24 °C) in the dark. We chose to focus on body skin rather than ear, snout and paws because the mouse body skin is more similar, although thinner, to human body skin at common sites of sand fly bite (forearms and lower legs). Afterwards, mice were removed from cages and administered 0.2 mg revontor to aid recovery in thermal cages at 37 °C. Sand flies were maintained in cages at 26 °C 65% humidity and 18/6 h light/dark photoperiods[35]. Fed sand flies were kept for 7 days to allow parasite development and expansion prior to analysis of parasite load.

**Determination of parasite load by impression smear.** Small pieces of mouse spleen (~10 µg) and liver (~25 µg) were cut and pressed repeatedly onto glass slights to generate impression smears. These were immediately fixed with

100% methanol, stained with 10% Giemsa in 67 mM Sorensen's Buffer (Na$_2$HPO$_4$/KH$_2$PO$_4$, pH 6.8) for 20–30 min, rinsed briefly with water and air dried[36]. 1,000 mouse cell nuclei were counted per organ and the corresponding parasites per field of view. Parasite numbers per 1,000 mouse nuclei were multiplied by the organ weight (g) to derive Leishman Donovan Units (LDUs)[37].

**Genomic DNA extraction**. All genomic DNA (gDNA) extractions were performed with either the DNeasy Blood & Tissue spinning column kit (Qiagen) or the DNeasy 96 Blood & Tissue plate kit (Qiagen) according to supplier's protocol. Per mouse, 100 µl of blood and 24 punch biopsies (ø 0.8 cm) were collected from the body skin (excluding paws, snout and ears) in a regular pattern (Fig. 1b) for gDNA extraction. Per fly exposure, 40 fed sand flies were collected per mouse for gDNA extraction.

**Quantitative polymerase chain reaction**. *Leishmania*-specific kinetoplastid DNA primers used in this study were previously characterized by Bezerra-Vasconcelos et al. (2011) (Accession number AF103738)[38] and used at a final concentration of 200 nM. 2 ng of gDNA were used per reaction. Fast SYBR Green Master Mix (Applied Biosystems) was used according to supplier's guidelines. Reactions were run in a StepOnePlus thermal cycler (Applied Biosystems) with a thermal cycle of 95 °C for 20 s, a cycling stage of 40 cycles of 95 °C for 3 s, 60 °C for 5 s, 72 °C for 20 s, 76.5 °C for 10 s (data read at final step), followed by the standard melt curve stage. Data were analyzed by StepOne Software v.2.3.

**Lymphocyte preparation and adoptive transfer**. T-cell extraction from spleen tissue was adapted from Owens et al.[39]. Briefly, spleens from naive male C57BL/6 mice (8–14 weeks of age) were forced through a 70 µm pore tissue strainer, spun down at $306 \times g$ for 5 min at 4 °C in a Heraeus Multifuge 3$_{S-R}$, cell pellets suspended in 5 ml ACK lysing buffer, incubated for 5 min at room temperature, spun down as before, washed in RPMI and spun down again as before. Cells were resuspended and counted with Trypan Blue under a light microscope. Isolated splenocytes were either fluorescently labeled by monoclonal CD45 (103108; 1:100), CD3 (100228; 1:100), CD4 (100412; 1:100), CD8 (126610; 1:100) antibodies (BioLegend) and submitted to MoFlo cytometry or labeled with CD4 (130-049-201) or CD8 (130-049-401) MicroBeads attached with antibodies diluted according to supplier's guidelines (1:10 per 10$^7$ splenocytes; Miltenyi Biotec) and passed twice through MACS LS columns (Miltenyi Biotec) according to supplier's guidelines for optimal T-cell purification. 10$^6$ CD4$^+$ or CD8$^+$ T cells were injected into infected RAG mice via the lateral tail vein according to the schedule described.

**Microscopy**. For fluorescent stereomicroscopy, whole skins from tdTom-*L. donovani*-infected B6.CD45.1*.Rag2$^{-/-}$* mice were harvested and the excess hypodermal adipose tissue scraped off. Skins were imaged from the hypodermal face to visualize tdTom-*L. donovani*. A series of 40-50 images were taken at 12× magnification to cover the whole skin. Images were aligned and merged in Photoshop to render a single whole skin image. No further image manipulation was performed.

For confocal microscopy, punch biopsies (ø 0.8 cm) of infected mouse skin were fixed in 4% Formaldehyde for 3–4 h, immersed in 15% sucrose solution for 12 h and 30% sucrose solution for 24 h at 4–8 °C prior to embedding in OCT and freezing on dry ice. Frozen samples were preserved at −80 °C. Frozen skin samples were longitudinally sectioned at 10 µm thickness to expose all skin layers and placed on microscopy glass slides. Skin sections were rehydrated for 5 min in PBS + 0.05% BSA and serum, and washed once in PBS. Sections were stained with DAPI (1 µg ml$^{-1}$) in PBS for 5 min at room temperature in the dark. They were then washed twice in PBS, mounted in ProlongGold mounting media and stored at 4 °C overnight. Slides were sealed the next day and imaged with a 40× oil-immersion objective (400 × magnification) on an upright confocal microscope with a 710 Metahead (Zeiss) by z-stack tile-scans.

**Definition of volumes and sizes**. All parameters with values used in the model are listed in Supplementary Table 3. In the following, we will detail how the parameters were derived.

Skin (punch) biopsies had a diameter of 0.8 cm restricted by puncher diameter. Sabino et al. measured male C57BL/6 mouse skin to be ~240 µm (epidermis: 12.53 ± 3.97 µm; dermis: 226.33 ± 36.86 µm)[40]. The skin biopsy volume was calculated by the equation for cylindrical volumes: $V = r^2\pi h = 4^2$ mm$^2 \times 3.14159 \times 0.24 \approx 12$ mm$^3$

Blood pools in the dermis can by visualized by imaging the skin from the hypodermal face. Diameter measurements for visible blood pools lay between 0.5 to 1.5 mm (Supplementary Fig. 21B). For the fixed blood pool volume, we, therefore, used 1 mm as the standard diameter. Since blood pools are limited on two sides (epidermis and hypodermis) from expanding, we assumed that blood pools could only expand cylindrically. Therefore, we used again the cylindrical volume equation to determine blood pool volumes: $V = r^2\pi h = 0.5^2$ mm$^2 \times 3.14159 \times 0.24 \approx 0.19$ mm$^3$. For variable blood pool volumes, we generated, for each sand fly, an individual blood pool with a uniform randomly assigned diameter

(0.5–1.5 mm) and, therefore, uniform random volume in the expanded model (0.05–0.42 mm$^3$).

Labrum length (~320 µm) and width (~29 µm) as measured by Brinson et al.[41] were used to describe the skin penetrating part of the *Lu. longipalpis* proboscis. We assumed a cylindrical shape of the proboscis, and that only about half of it penetrated the skin to calculate the proboscis volume: $V = r^2\pi h = 0.0145^2$ mm$^2 \times 3.14159 \times 0.16 \approx 0.000106$ mm$^3$

Maximum sand fly feed volume of 1.6 mm$^3$ for *Lu. longipalpis* is based on measurements from Rogers et al.[20], while feed volume range (0.1–1.6 mm$^3$) included measurements from Ready[42]. For variable feed volumes, we assigned each sand fly an individual uniform random feed volume in the expanded model.

There is no published natural outward transmission data available that previously established minimum amastigote infection thresholds. Anjili et al. reported previously a ~33% infection potential in artificially fed and fully engorged sand flies ingesting at least 10 amastigotes[22]. Using our data from the mice shown in Fig. 1, we modeled the infection threshold by gradually reducing amastigote infectivity in silico (Supplementary Note 4). The results showed that a ~33% infection potential is reached at a threshold requiring the ingestion of 1000 amastigotes. Using variable parameters allowed a reduction of the infection threshold to 500 amastigotes, closer to the 600 amastigotes estimated to be required for reliable sand fly infection using artificial membrane feeding[21].

**Statistical analysis and experimental design**. With no available published data on this topic at the time of study conception, it was not possible to perform prospective power calculations. We used $N = 9$ mice per group in the pilot study to have an $N$ and degree of freedom (df = 8 per group) at the mouse level sufficient for robust statistical analysis even with larger than expected data variance. Mouse experiments were repeated three times with an $N = 3$ per group per repeat (total $N = 9$ per group). To minimize impact of external factors on data variability, all mice were of the same gender (male) and maintained under the same conditions in accordance to Home Office guidelines. Power analyses assuming 90% power and $P < 0.05$ as significant with G*Power[43] post data collection suggested that a total sample sizes of 88 skin biopsies with four groups (total of 864 biopsies analyzed in this study) and 166 sand flies with three groups (1080 sand flies analyzed in this study). An $N = 9$ mice per group was sufficient to address the issue of inner data bias due to multiple samples taken per mouse (24 skin biopsies and 40 sand flies analyzed per mouse).

All statistics were performed using GraphPad PRISM v.5, SPSS v.23 or STATA v.14. We considered in all instances $P < 0.05$ to be significant and $P$-values are displayed to the third decimal position. All datasets were tested in SPSS for normal distribution by Shapiro-Wilks test, Q-Q plots and histograms and for homogeneity of variance by Brown-Forsythe's test. Skin biopsy data lacked homogeneity of variance and sand fly data was strongly right skewed due to the inherent zero bias. Data were analyzed by the Box-Cox procedure in SPSS to identify optimal data transformation to normality, where possible. In general, data were most efficiently transformed by log$_{10}$ transformation. Where parametric assumptions were met, data sets were analyzed by un-/paired $t$-test or analysis of variance (ANOVA) followed by Bonferroni and Tukey post hoc tests. Otherwise, analysis was done by Mann-Whitney $U$ test, Wilcoxon signed-rank test or Kruskal-Wallis test. Parasite ratios across skins for the 24 biopsy sites and the comparison of model and experimental data were analyzed by Friedman test with a Wilcoxon signed-rank test as a *posthoc* analysis. We used simple and multiple linear regression analyses in SPSS v.23 and STATA v.14 to investigate independent predictors of parasite loads in sand flies. Data residuals were checked for normal distribution to verify test validity. Variables showing a $P < 0.05$ were considered to be significant predictors. We used log$_{10}$-transformed mean parasite numbers in sand flies, skin, blood, spleen and liver in the regression analyses.

For experimental randomization and blinding, bar-coded microchips were implanted in the tail of each mouse used in this study. For treatment, mice were selected at random and microchips were read prior to applying the required treatment as per protocol (and without reference to any previous treatments). The key to the bar code was only revealed post conclusion of all experimental analyses.

**Mathematical models**. The mathematical models are based on established descriptions of predators foraging in patchy environments[16, 17], adapted here to allow for patchiness at multiple scales.

The simplest preliminary model (Supplementary Fig. 3C–E) assumed that each sand fly feeds at a single location and ingests a fixed volume of blood V$_{FF}$. Parasites are ingested according to a Poisson process with a rate proportional to the local parasite density $\mu$. With a homogeneous distribution, all local parasite densities are equal, resulting in a Poisson (approximately normal) ingestion distribution with mean V$_{FF}$ $\mu$ (Supplementary Fig. 3C). In contrast, assuming that a low density of parasites is distributed randomly throughout the skin, but that there also exist local patches of increased parasite density (constrained so that the spatially-averaged parasite density remains $\mu$), resulting in a bimodal distribution where a small number of sand flies ingest a large number of parasites (Supplementary Fig. 3D; in these simulations it is assumed that 80% of the parasites are concentrated within patchiness that comprises 30% of the skin surface area).

The empirical data allow better characterization of this patchiness. For each mouse $i$ an NB($\mu_i$, $k_i$) is fitted. For each sand fly feeding on mouse $i$, an

independent NB($\mu_i$, $k_i$), random variable $Z$ is then generated, and the number of parasites ingested by that sand fly is then simulated as a Poisson random variable with mean $V_{FF}\,\mu_i$. This mixture-based approach allows the role of patchiness (quantified here by the $k_i$) to be evaluated and manipulated (Supplementary Fig. 3E).

The results presented in Fig. 5 and Supplementary Figure 16 extend these concepts to account for patchiness or homogeneity (P or NP) at both the macro- (Ma) and micro- (Mi) scales. In the MaPMiP case (i.e., patchiness at both scales) for each fly $j$ feeding on mouse $i$ an independent NB($\mu_i$, $k_i$) random variable $Z_i$ again characterizes local macroscopic parasite density. Micro-scale patchiness is then modeled by a further mixture: the number of ingested parasites is a NB($Z_i$, $k_{Micro}$) random variable (Supplementary Table 2) where $k_{Micro}$ quantifies the micro-scale heterogeneity and the feeding at the micro-scale is assumed to be a Poisson process with a rate proportional to the local parasite density $\mu$ at the micro site.

**Code availability.** Source codes and documentation for all computational models are provided in Supplementary Software 1.

**Data availability.** The authors declare that the data supporting the findings of the study are available within the article and its Supplementary Information files, or are available from the authors upon request.

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

## Acknowledgements

This work was supported by a Wellcome Senior Investigator Award (# WT106203; http://www.wellcome.ac.uk) and a MRC Programme Grant (#G1000230; https://www.mrc.ac.uk) to P.M.K. S.D. and H.D. were supported by the Wellcome Trust-funded PhD program "Combatting Infectious Diseases, Computational Approaches in Translational Science (http://www.york.ac.uk/cidcats/). The authors thank Ian Graham for commenting on the manuscript, Piotr Naskrecki for sand fly images used in Fig. 4, the staff of the Biological Services Facility at the University of York for animal husbandry and support and staff at Fera Science Ltd for sand fly husbandry.

## Author contributions

J.S.P.D., P.M.K. and J.W.P. designed the study; J.S.P.D., N.B., A.R., J.E.D. and A.I.P. conducted the experimental studies; Z.B., S.D. and H.D. developed the mathematical models; J.M. developed, maintained and improved the sand fly colony; J.S.P.D., P.M.K. and J.W.P. wrote the manuscript and all authors contributed to the final version; J.W.P. and P.M.K. supervised the study and share equal senior authorship; P.M.K. conceived the study.

## Additional information

**Competing interests:** The authors declare no competing financial interests.

