## [Peer Review File · Nature Communications]

Reviewers' comments:

Reviewer #1 (Remarks to the Author):

I was intrigued by the original paper and found it interesting. I was puzzled by the lack of specificity with which the word model was used in the paper. After hearing back from the authors I read the background paper by Pitchford and Brindley in the Bulletin of Mathematical Biology (BMB) (2001) which helped clarify what I think needs to be added to the paper, mostly in the supplementary materials. The BMB paper actually contains two models, one a derivation of a predator-prey encounter rate in a patchy environment and the other a model for larval growth. To my way of thinking, I would probably call the encounter rate a term in a model rather than a model per se, but that may be quibbling on my part. However, I think a sketch of the overall modeling framework needs to be laid out somewhere. That can certainly be in the supplemental materials. That way when the term model is brought up in the main exposition the reader can have some clarity about exactly what is being discussed. Particularly curious readers can then explore MATLAB for more detail.

Reviewer #2 (Remarks to the Author):

This is a review of "Skin parasite landscape determines host infectiousness in visceral leishmaniasis". The authors run a multidisciplinary study, which combines laboratory work, statistical analysis and mathematical modeling, which provides for the first time strong quantitative evidence for the skin infectiousness of visceral leishmaniasis (VL) using a mouse model. The importance and novelty of the study stems from the fact that this work demonstrates the potential and relative importance of skin parasites in infecting sand flies. The findings of this manuscript are of true relevance since the London Declaration on Neglected Tropical Diseases, as a coordinated effort among international agencies, pharmaceutical companies, and other health authorities, has established the goal of eliminating VL as a public health problem by 2020. However, at present, over 500,000 new cases of VL are reported worldwide each year and the disease is second only to malaria among vector-borne infectious diseases in terms of attributable mortality. Despite implementation of novel case-detection strategies, rapid diagnostic testing, and vector control activities, attainment of the elimination goal is unlikely due to complex transmission dynamics, struggling health systems, and poor diagnosis rates, among other factors. Identifying criteria for assessing risk of infection would allow more effectively targeted intervention to prevent transmission. Moreover, as the authors point out in the manuscript it seems that blood parasitemia is only a partial indicator of the transmission potential of VL and there is need for more data and research to understand the complex transmission of VL. As leishmaniasis is a skin parasite it is not surprising that the skin can harbor the parasite; this combined with the fact that the ratio between infection and disease is about 1:20 suggest that asymptomatic hosts play a vital role in the ecology and transmission dynamics of VL. Previous studies (as mentioned in the paper) have pointed out the potential importance of skin parasites to transmission of VL. The significance of the current paper is that by conducting intensive laboratory work including an extensive Xenodiagnosis study,

which is a particular difficult experiment, unique and valuable data was created. By conducting a rigorous statistical analysis the authors have established and quantified an empirical basis between skin parasite load and infection of sand flies. The statistical study of the data indicated that the parasites are distributed heterogeneously in the skin. In the third step, a mathematical model was developed to investigate and further quantify the significance of the parasite heterogeneity on the transmission of parasites to sand-flies. The modeling suggests that both heterogeneities in the micro and macro scales are important to reproduce the experimental results.

I would like to see more details regarding the mathematical model. In the current form of the paper, it is very difficult to reproduce the modeling results. Please provide the equations used and a table of the parameters, their meanings and values. I would also suggest conducting a sensitivity analysis of the model. Please try and better identify the conditions in which parasite heterogeneity is important. Currently there is a general discussion of this issue but more details (e.g., the conditions as a function of the model parameters where heterogeneity enhances transmission) will be crucial to assess how important heterogeneity is. As the authors write, the output of the model is dependent on the threshold value. I believe that (as written in the manuscript) the results will not change in a qualitative way by changing the threshold value, nevertheless I think adding a figure in the supplementary information demonstrating this will be important. In addition, what will be the results if there is no threshold?

The authors demonstrate that the negative binomial distribution is a good description of the parasites in the skin and use their findings in the simulations. I would suggest that in addition to the negative binomial distribution additional simulations can be run by bootstrapping the empirical parasite distributions found in the mice. I believe this will be a complementary empirical approach, which will aid in corroborating the findings without the need to make assumptions regarding the parasite distribution.

I would like to point out the fact that one of the major findings here is that heterogeneous transmission enhances disease transmission; this is in accordance with previous studies which found similar phenomena at the population level (in the case of vector-borne diseases including VL) see for example Dye 1986, 1988, Smith 2004 PloS Biol, Miller 2013, Miller 2016.

Additional points which require further attention:

1. The authors mentioned feeding flies on the shaved skin on the side of mice. It is not clear what area (how large) of skin was available to the flies, whether it was marked and later checked by fluorescence to determine if it was heavily infected, not infected, or whether parts were and parts were not (perhaps this is clarified in other sections but I didn't see it in materials).
2. If allowed to select freely (anaesthetized mouse inside large cage), most flies would feed on the snout, around the mouth, on the ears and on the feet (these areas have only sparse fur allowing access). Were these parts analyzed for parasitemias?
3. Previous works (on dogs and mice) tend to indicate heavier parasitemias close to the inoculation site by the infectious sand fly bite. Sandflies inoculate the parasites into the

dermis. The authors infected their mice intravenously. This is probably the reason they are not seeing localization close to the infection site.

4. Had they infected the mice by intradermal injection or using infected sand flies, my guess is they would have localized infections on the nose ears (or close to the (intradermal) injection site)

5. I think that skin infections should be characterized as either primary = at the site of infection by sand fly bite (as in all cutaneous leishmaniasis and the two papers they cite in mice – Seblova et al and in dogs – Aslan et al) or secondary evolving after cure of VL = e.g. PKDL in humans. The results of this study appear more like secondary infections because they arise from IV injection

Given the importance of this study, I believe that the authors need to write the paper in a more clear and compact way. In the current form of the paper, there is too much detail and information in the main text, which make it very difficult to follow. I suggest moving several of the figures and analysis to the supplementary information. Specifically for figure 1 there are 10 subplots; I would move at least 4 to the supplementary information. Please keep the subplots which are crucial to understanding the study. In figure 2, which has 8 subplots, I would move the figure with the CD4 and CD8 to the supplementary information. The reader can understand the main points by referring to the fly-fed data. In figure 3 (12 subplots) I would show the 4 most significant correlations in the main text. I would move the entire figure 4 to the supplementary information, it gives the reader very little insight. I would also move figures 6 & 7 to the supplementary information.

I believe that this study is an excellent example of combining experimental work with statistics and modeling to obtain new insight into a complex biological system. Moreover, as I wrote above, this work also has direct implications on public health and should be used to develop in the future improved mitigation studies to reduce the burden of this important disease. I look forward to seeing a revised manuscript.

Reviewer #3 (Remarks to the Author):

I enjoyed reading this manuscript and found the results and conclusions well supported, and an important contribution to understanding the transmission dynamics of visceral leishmaniasis. The only comments I would add, and suggest the authors modify the manuscript accordingly, is to note more explicitly the pool-feeding behaviour of the sand fly vector. This is implied but not described, yet this crucial feature that distinguishes sand flies from say mosquitoes, makes the results of the study considerably more important than might be thought otherwise, even predictable. Personally I have always regarded *Leishmania* as parasites of the skin that occasionally occur elsewhere, so the results of the manuscript are of no great surprise to me, although I suspect that point has escaped most working on this disease that are not familiar with vector behaviour. The destructive feeding behaviour of sand flies, which cause tissue damage in the skin and then feed from the

resulting pool of blood is very different to the capillary seeking behaviour of mosquitoes. If blood parasitaemia was the crucial driver of outward transmission then natural selection would not have favoured sand flies as vectors, something like a mosquito would be the vector. Hence the converse is true, the skin parasitaemia and distribution, as shown here, is consistent with a pool feeding vector. Thus in a way vector biology predicts these findings, of course they have to be demonstrated, as elegantly done in this study. Arguments such as these can be used to strengthen the conclusions of this work.

Doehl et al NCOMMS-16-30280 Response to reviewers

Reviewer #1

“I was puzzled by the lack of specificity with which the word model was used in the paper. After hearing back from the authors I read the background paper by Pitchford and Brindley in the Bulletin of Mathematical Biology (BMB) (2001) which helped clarify what I think needs to be added to the paper, mostly in the supplementary materials. The BMB paper actually contains two models, one a derivation of a predator-prey encounter rate in a patchy environment and the other a model for larval growth. To my way of thinking, I would probably call the encounter rate a term in a model rather than a model per se, but that may be quibbling on my part. However, I think a sketch of the overall modeling framework needs to be laid out somewhere. That can certainly be in the supplemental materials. That way when the term model is brought up in the main exposition the reader can have some clarity about exactly what is being discussed. Particularly curious readers can then explore MATLAB for more detail.”

We have taken on board the reviewers comments and modified the text so that the term “model” is used now in only two contexts. The first is the biological one, namely the experimental mouse and sand fly transmission model. The second is the mathematical model, which analyses macro- and micro-scale patchiness. We have avoided the use of this term elsewhere. The Supplementary Materials now provides an expanded description of the mathematical model, as requested (**new text: Supplementary Text S1**).

Reviewer #2

“I would like to see more details regarding the mathematical model. In the current form of the paper, it is very difficult to reproduce the modeling results. Please provide the equations used and a table of the parameters, their meanings and values.”

We agree that the original manuscript was light on detail of the mathematical modelling. As requested, we have now expanded the description of the modelling, providing all the information requested by the reviewer (**new text: Supplementary Text S1.1**).

“I would also suggest conducting a sensitivity analysis of the model.”

We have considered the reviewers request for a sensitivity analysis, but in our view, including a full sensitivity analysis would be inappropriate. The output of such a process would be ambiguous because we would be assessing changes in fit to distributional data, and comparing these across different treatments, rather than assessing the change in a single predicted number. Moreover, the process would need to be repeated using different assumptions about the role of patchiness (micro-, macro- or both.) Instead, the structural sensitivity of the modelling framework is explored by systematic consideration of patchiness at different scales. In this sense, we argue that the data presented in Figure 5 are more effective in highlighting the important message of the manuscript.

“Please try and better identify the conditions in which parasite heterogeneity is important. Currently there is a general discussion of this issue but more details (e.g., the conditions as a function of the model parameters where heterogeneity enhances transmission) will be crucial to assess how important heterogeneity is. As the authors write, the output of the model is dependent on the threshold value. I believe that (as written in the manuscript) the results will not change in a qualitative way by changing the threshold value, nevertheless I think adding

a figure in the supplementary information demonstrating this will be important. In addition, what will be the results if there is no threshold?”

We have now provided more information on the issue of the threshold as suggested by the reviewer (**new text: Supplementary Text 1.2.2; new figure: Supplementary Fig. 19**). This new analysis does not substantively affect the conclusions of the manuscript.

“The authors demonstrate that the negative binomial distribution is a good description of the parasites in the skin and use their findings in the simulations. I would suggest that in addition to the negative binomial distribution additional simulations can be run by bootstrapping the empirical parasite distributions found in the mice. I believe this will be a complementary empirical approach, which will aid in corroborating the findings without the need to make assumptions regarding the parasite distribution.”

We thank the reviewer for this suggestion and have performed bootstrapping as requested (**new text: Supplementary Text S1.2.3; new figure: Supplementary Fig. 20**). As above, this new analysis complements but does not affect the conclusions of the manuscript.

“I would like to point out the fact that one of the major findings here is that heterogeneous transmission enhances disease transmission; this is in accordance with previous studies which found similar phenomena at the population level (in the case of vector-borne diseases including VL) see for example Dye 1986, 1988, Smith 2004 PloS Biol, Miller 2013, Miller 2016.”

We agree with the reviewer and have now included a new paragraph on population level heterogeneity, to indicate the similarities with our analysis (including reference to the work highlighted by the reviewer) (**new text: lines 365-379; Refs 8, 24-26**).

Additional points which require further attention:

1. *“The authors mentioned feeding flies on the shaved skin on the side of mice. It is not clear what area (how large) of skin was available to the flies, whether it was marked and later checked by fluorescence to determine if it was heavily infected, not infected, or whether parts were and parts were not (perhaps this is clarified in other sections but I didn’t see it in materials).”*

We have now added more detail of the procedures to address these points (**new text: lines 451-453; new Supplementary Figure 21**). In brief, the area exposed for sand fly bite is equivalent to approx. 8-10 punch biopsies. It is not possible to image for parasite load before feeding or after feeding (as the flies destroy the tissue where they feed and engorge any associated parasites). Hence, we have assumed that the patchiness identified in skin applies across the area of skin that flies are exposed to.

2. *“If allowed to select freely (anaesthetized mouse inside large cage), most flies would feed on the snout, around the mouth, on the ears and on the feet (these areas have only sparse fur allowing access). Were these parts analyzed for parasitemias?”*

We have now made it clear that these areas were excluded and included our rationale for this (new text: lines 456-458). Briefly, we focused on flank skin as this most closely approximates human skin in structure and thickness.

3. *“Previous works (on dogs and mice) tend to indicate heavier parasitemias close to the inoculation site by the infectious sand fly bite. Sandflies inoculate the parasites into the dermis. The authors infected their mice intravenously. This is probably the reason they are not seeing localization close to the infection site.”*

We agree with the reviewer on this point, but do not feel that it affects the conclusions drawn in this manuscript.

4. *“Had they infected the mice by intradermal injection or using infected sand flies, my guess is they would have localized infections on the nose ears (or close to the (intradermal) injection site).”*

Again, we agree with the reviewer that this is likely to be the case early after infection, but do not feel that it affects the conclusions drawn in this manuscript. However, our model addresses transmission potential of the skin after an extensive period of systemic disease and we believe that further dissemination to multiple skin sites is likely at this time (see below).

5. *“I think that skin infections should be characterized as either primary = at the site of infection by sand fly bite (as in all cutaneous leishmaniasis and the two papers they cite in mice – Seblova et al and in dogs – Aslan et al) or secondary evolving after cure of VL = e.g. PKDL in humans. The results of this study appear more like secondary infections because they arise from IV injection.”*

We agree and like the reviewer’s classification of primary and secondary skin infections. We have now introduced the biological model by referring to this classification (**new text: lines 91-94**).

“Given the importance of this study, I believe that the authors need to write the paper in a more clear and compact way. In the current form of the paper, there is too much detail and information in the main text, which make it very difficult to follow. I suggest moving several of the figures and analysis to the supplementary information. Specifically for figure 1 there are 10 subplots; I would move at least 4 to the supplementary information. Please keep the subplots which are crucial to understanding the study. In figure 2, which has 8 subplots, I would move the figure with the CD4 and CD8 to the supplementary information. The reader can understand the main points by referring to the fly-fed data. In figure 3 (12 subplots) I would show the 4 most significant correlations in the main text. I would move the entire figure 4 to the supplementary information, it gives the reader very little insight. I would also move figures 6 & 7 to the supplementary information.”

We thank the reviewer for the valuable suggestions on how to improve the readability of the manuscript and have made all of the changes to the text and Figures suggested above except one (**Figures 1-5 revised and new or revised Supplementary Figures 3, 8, 9, 15, 16, 17**). We have retained Figure 4 in the main manuscript as we feel that it does provide a good visual summary of the mathematical modelling. To accommodate the changes to the Figures and order of Figure call out, we have revised the layout of the text at various places throughout the manuscript (as indicated in marked copy).

Reviewer #3

“The only comments I would add, and suggest the authors modify the manuscript accordingly, is to note more explicitly the pool-feeding behaviour of the sand fly vector. This is implied but not described, yet this crucial feature that distinguishes sand flies from say mosquitoes, makes the results of the study considerably more important than might be thought otherwise, even predictable. Personally I have always regarded Leishmania as parasites of the skin that occasionally occur elsewhere, so the results of the manuscript are of no great surprise to me, although I suspect that point has escaped most working on this

disease that are not familiar with vector behaviour. The destructive feeding behaviour of sand flies, which cause tissue damage in the skin and then feed from the resulting pool of blood is very different to the capillary seeking behaviour of mosquitoes. If blood parasitaemia was the crucial driver of outward transmission then natural selection would not have favoured sand flies as vectors, something like a mosquito would be the vector. Hence the converse is true, the skin parasitaemia and distribution, as shown here, is consistent with a pool feeding vector. Thus in a way vector biology predicts these findings, of course they have to be demonstrated, as elegantly done in this study. Arguments such as these can be used to strengthen the conclusions of this work.”

We thank the reviewer for these insights, which we agree with entirely. We have modified the text in the introduction and discussion to make stronger reference to the pool feeding behaviour of sand flies and some of the arguments made above (**new text: lines 61-65, 112-114 and 400-401**).

REVIEWERS' COMMENTS:

Reviewer #1 (Remarks to the Author):

The revision has brought in enough details about the modeling in the pair so that the paper can be understood as a stand alone entity, which was the main thing that I thought needed to be done.

Reviewer #2 (Remarks to the Author):

The authors have done a nice job on revising the paper. They have followed most of my suggestions.

I have a few small remarks.

- 1) I share with the authors the philosophy that their model is a strategic model and it should be kept as simple as possible. In the revised manuscript they do a good job at explaining the model. I still do believe that a sensitivity analysis has a value here, due to the fact that the model is fitted to data which implies it has predictive potential. Nevertheless, I think the amount of detail in the paper and supplementary material is enormous which is sufficient in my view for publication (it is very impressive). I leave it to the editor to decide about the sensitivity analysis.
- 2) Regarding the infection threshold (figure 16 SI), my comment about a model without a threshold was misunderstood. What I implied by no threshold is that one should use a model where the probability of infection is proportional to the infection level.
- 3) I think there are a few typo's along the manuscript for example bottom of page 11 fig2c should be fig 3c.
- 4) One additional point which might be useful for the authors to think about is vector preference, if the vector has preferred biting sites (e.g., exposed skin areas in humans or ears in dogs and mice) this can enhance transmission as well (see Miller & Huppert PloS one 2013 & Miller et al 2016 Interface). I am not suggesting the authors cite these papers (it is not necessary) or revise the manuscript to include this point!

Over all this is excellent paper and I am looking forward to see it published

Response to reviewers' comments

Reviewer #1

No further revisions required.

Reviewer #2

The authors have done a nice job on revising the paper..... Over all this is excellent paper and I am looking forward to see it published.

Response: We thank the reviewer for his/her very positive comments .

I have a few small remarks.

1) I share with the authors the philosophy that their model is a strategic model and it should be kept as simple as possible. In the revised manuscript they do a good job at explaining the model. I still do believe that a sensitivity analysis has a value here, due to the fact that the model is fitted to data which implies it has predictive potential. Nevertheless, I think the amount of detail in the paper and supplementary material is enormous which is sufficient in my view for publication (it is very impressive). I leave it to the editor to decide about the sensitivity analysis.

Response: We thank the Editor for indicating that a sensitivity analysis is not required for publication.

2) Regarding the infection threshold (figure 16 SI), my comment about a model without a threshold was misunderstood. What I implied by no threshold is that one should use a model where the probability of infection is proportional to the infection level.

Response: We apologise that we had misunderstood what the reviewer requested. The best available empirical data suggest a minimum of 600 parasites is required for reliable infection, and our results implement this as a simple threshold. The referee is correct that we could revise this by adding a more complicated probabilistic model, for example

$$P(\text{infection} \mid i \text{ parasites are ingested}) = i^p / (k^p + i^p)$$

where k represents a half-saturation constant (presumably in the range 0-600) and the exponent p reflects the degree of nonlinearity (so $p = 1$ is locally linear for small i , and our threshold model is approached as p tends to infinity). Another approach, which is perhaps more interesting and more biologically realistic, is to apply individual-level variability in the value of the simple threshold (motivated by the Lloyd-Smith 2005 models).

We have implemented both these model refinements in exploratory simulations, and they show broadly similar results. However, there is no evidence in favour of either of these approaches. Moreover, either of these refinements requires the identification and estimation of additional parameters for which we do not have data. Therefore, we prefer with the editor's permission to limit our results to those involving a simple, and empirically defensible, threshold.

3) I think there are a few typo's along the manuscript for example bottom of page 11 fig2c should be fig 3c.

Response: We have thoroughly re-checked the manuscript and associated files and corrected any typographical / grammatical errors.

4) One additional point which might be useful for the authors to think about is vector preference, if the vector has preferred biting sites (e.g., exposed skin areas in humans or ears in dogs and mice) this can enhance transmission as well (see Miller & Huppert PloS one 2013 & Miller et al 2016 Interface). I am not suggesting the authors cite these papers (it is not necessary) or revise the manuscript to include this point!

Response: We thank the reviewer for making this point and of course acknowledge that biting site preference may also impact on transmission potential at different sites. We feel that even taking this into account, the extent of micro- and macro- scale heterogeneity will be important. As indicated by the reviewer, we have not made changes to the manuscript to specifically address this point .